# Optimising the Polyphenolic Content and Antioxidant Activity of Green Rooibos (*Aspalathus linearis*) Using Beta-Cyclodextrin Assisted Extraction

**DOI:** 10.3390/molecules27113556

**Published:** 2022-05-31

**Authors:** Lusani Norah Vhangani, Leonardo Cristian Favre, Guido Rolandelli, Jessy Van Wyk, María del Pilar Buera

**Affiliations:** 1Department of Food Science & Technology, Cape Peninsula University of Technology, Bellville 7535, South Africa; vanwykj@cput.ac.za; 2CONICET—INTA, Instituto de Ciencia y Tecnología de los Sistemas Alimentarios Sustentables (ICyTeSAS), Las Cabañas y De Los Reseros s/n, 1686, Buenos Aires C1425FQB, Argentina; cristihanfav@gmail.com; 3CONICET—Consejo Nacional de Investigaciones Científicas y Técnicas, Godoy Cruz 2290, Buenos Aires C1425FQB, Argentina; rolandelliguido@gmail.com (G.R.); pilar.buera@gmail.com (M.d.P.B.); 4Departamento de Industrias y Departamento de Química Orgánica, Facultad de Ciencias Exactas y Naturales, Universidad de Buenos Aires, Intendente Güiraldes 2160, Ciudad Universitaria, Buenos Aires C1428EGA, Argentina; 5Instituto de Tecnología de Alimentos y Procesos Químicos (ITAPROQ), CONICET-Universidad de Buenos Aires, Intendente Güiraldes 2160, Ciudad Universitaria, Buenos Aires C1428EGA, Argentina

**Keywords:** green rooibos, β-cyclodextrin extraction, antioxidant activity, polyphenolic content, encapsulation

## Abstract

Antioxidant activity associated with green rooibos infusions is attributed to the activity of polyphenols, particularly aspalathin and nothofagin. This study aimed to optimise β-cyclodextrin (β-CD)-assisted extraction of crude green rooibos (CGRE) via total polyphenolic content (TPC) and antioxidant activity assays. Response surface methodology (RSM) permitted optimisation of β-CD concentration (0–15 mM), temperature (40–90 °C) and time (15–60 min). Optimal extraction conditions were: 15 mM β-CD: 40 °C: 60 min with a desirability of 0.985 yielding TPC of 398.25 mg GAE·g^−1^, metal chelation (MTC) of 93%, 2,2′-azino-bis(3-ethylbenzothiazoline-6-sulfonic acid) (ABTS) radical scavenging of 1689.7 µmol TE·g^−1^, ferric reducing antioxidant power (FRAP) of 2097.53 µmol AAE·g^−1^ and oxygen radical absorbance capacity (ORAC) of 11,162.82 TE·g^−1^. Aspalathin, hyperoside and orientin were the major flavonoids, with quercetin, luteolin and chrysoeriol detected in trace quantities. Differences (*p* < 0.05) between aqueous and β-CD assisted CGRE was only observed for aspalathin reporting the highest content of 172.25 mg·g^−1^ of dry matter for extracts produced at optimal extraction conditions. Positive, strong correlations between TPC and antioxidant assays were observed and exhibited regression coefficient (R^2^) between 0.929–0.978 at *p* < 0.001. These results demonstrated the capacity of β-CD in increasing polyphenol content of green rooibos.

## 1. Introduction

The indigenous South African plant species *Aspalathus linearis*, traditionally known as the rooibos plant, grows naturally in the Fynbos biome in the Cederberg area in the Western Cape Province of South Africa [1]. *Aspalathus linearis* has been deemed an invaluable traditional medicine. Interviews conducted with indigenous Khoi-San communities relating to the medicinal use of rooibos revealed that it had been used to manage high blood pressure, aid digestion, treat eczema and allergies and stimulate appetite, among other uses [2]. Rooibos is commercialised in two different forms: unfermented and fermented rooibos, with the former, reported to exhibit higher antioxidant activity due to higher quantities of aspalathin and nothofagin [3,4].

Besides, *Aspalathus linearis* infusions have received significant attention in recent years due to their bio-functional properties, with numerous studies having been conducted on health-promoting properties associated with rooibos. These include antioxidant [5], anti-inflammatory [6], antimicrobial [7], anti-obesity and hypoglycaemic [8] activities. The main compounds with bio-functionality are phenolic compounds [1], exhibiting their properties by inhibiting the oxidation of low-density lipoprotein (LDL) via radical scavenging [9]. Polyphenols are secondary plant metabolites involved in defence against ultraviolet radiation or attack by pathogens [10,11]. Polyphenols impact the organoleptic properties of foods, contributing to bitterness, astringency, colour, flavour, odour and oxidative stability [11].

Moreover, there is a globally growing market for plant bioactive compounds due to the functionalities mentioned above. Therefore, to fully exploit these bio-functional properties, emphasis must be placed on finding more novel techniques that maximise extraction yield from plant waste and by-products for potential use as nutraceuticals [12]. Extraction yield for the recovery of these antioxidants from plants and the obtained antioxidant activity depends on the extraction method and the solvent used for extraction. Water is the safest solvent that may be used to extract bioactive compounds from plant material; however, its capability is limited to polar compounds and results in lower extraction efficiency [13].

On the other hand, polar compounds are frequently used for recovering polyphenols from plant matrices. The most suitable solvents are aqueous mixtures containing ethanol, methanol, acetone or ethyl acetate [14]. Although these solvents effectively increase the extraction yield, enormous volumes are used, and they are not environmentally friendly by contributing to pollution. Hence “greener” extraction procedures are sought to reduce the environmental impact caused by these solvents [15,16].

Alternative environmentally friendly extraction processes, such as supercritical fluid extraction, pulsed electric fields and ultrasound, are considered to reduce organic solvent use and their safety concerns [17,18,19]. However, the cost associated with the procurement of these alternative technologies might prohibit their implementation. Cyclodextrins (CDs) are cyclic oligosaccharides with a truncated cone spatial geometry. They are widely used in the food, pharmaceutical and chemical industries for their ability to form host–guest inclusion complexes with a wide range of bioactive compounds [15,18]. Cyclodextrins result in modification of the encapsulated compound’s physicochemical properties, leading to improved solubility, stability [19] and bioavailability, as well as their application in separation and purification operations. Numerous studies have been conducted where the ability of CDs to improve the extraction of polyphenols from plant matrices was investigated. Extraction parameters, such as type of CDs and concentration, temperature, and time are likely to influence the process, having possible interactions among the variables [15,20,21,22]. For instance, the extraction of some phenolic compounds from plants with different aqueous CDs solutions has been demonstrated to be an efficient and eco-friendly extraction process [22]. A study by [20] proved that amongst α-CD, γ-CD and β-CD, the latter was the most effective in recovering stilbenes, flavonols and flavan-3-ols from grape pomace.

On the other hand, ref. [19] compared the efficacy of solvents, namely, water, methanol and hydro-ethanol, with that of the β-CD-water solution to extract polyphenols from vine shoots. Using the β-CD solution, the extraction time was reduced by half compared with pure water as a solvent. In addition, β-CD-assisted extraction also resulted in higher polyphenols content than hydro-ethanol [19]. Moreover, the encapsulation role of β-CD extracted polyphenols also increased stability against degradation by oxidation compared to methanolic extracts. Since the β-CD concentration, time and temperature must be defined, response surface methodology (RSM) helps select optimal extraction parameters that result in high yield and improved functional properties [23].

Rooibos tea is a unique product, and its production is implemented under sustainability programs, aiming to protect biodiversity and social developments. Beta-cyclodextrin assisted extraction can improve the antioxidant activity of green rooibos and contribute to diversifying these plant applications. The current study was carried out to obtain the β-CD concentration-time and temperature combination for the optimal recovery of polyphenols from green rooibos.

## 2. Materials and Methods

### 2.1. Green Rooibos and Reagents

The dry green rooibos was obtained from a major local producer (Rooibos Ltd., Clanwilliam, South Africa). Beta-cyclodextrin (β-CD) was purchased from Industrial Analytical (Kyalami, South Africa). Folin and Ciocalteu’s phenol reagent, gallic acid monohydrate, and 2,4,6-tri[2-pyridyl]-s-triazine, ascorbic acid, 2,2′-Azobis (2-methylpropionamidine) dihydrochloride, fluorescein disodium salt, Trolox, (2,2′-Azino-bis(3-ethylbenzothiazoline-6-sulfonic acid), ethanol, polyphenolic standards quercetin and luteolin were purchased from Sigma-Aldrich (Kempton Park, South Africa). Orientin, iso-orientin, vitexin, isovitexin, hyperoside and chrysoeriol were authentic reference standards of purity ≥ 95% purchased from Extrasynthese (Genay, France). 

Aspalathin was a gift from the Oxidative Stress Research Centre at the Cape Peninsula University of Technology (Bellville, South Africa) sourced from (Rooibos Ltd., Clanwilliam, South Africa). Ethylenediaminetetracetic acid, glucose, potassium hydrogen phosphate, absolute ethanol, ascorbic acid, potassium ferricyanide, trichloroacetic acid, potassium persulfate, sodium chloride, acetic acid, ferrozine, ferric chloride, ferrous chloride acetonitrile and sodium carbonate, sodium acetate, hydrochloric acid, acetone, glacial acetic acid, perchloric acid, potassium-peroxodisulphate, citric acid and potassium metabisulphite were purchased from Merck (Modderfontein, South Africa).

All chemicals used in this study were analytical grade, and chemical reagents were prepared according to standard analytical procedures. Prepared reagents were stored under conditions that prevented deterioration or contamination. The water used in this study was ultrapure water purified with a Milli-Q water purification system (Millipore, Microsep, Bellville, South Africa).

### 2.2. Solid-Liquid Extraction of Green Rooibos

Green rooibos, as received, was coarsely milled (Fritsch) using a sieve with an aperture of 0.2 mm. The extraction of green rooibos was performed based on the method of [14,19] with slight modifications. Green rooibos plant of 10 g and a 100 mL of 0, 7.5 and 15 mM β-CD (0.85 and 1.7%, respectively) aqueous solutions (1:10 (*w*/*v*)) ratio in a Schott bottle were homogenised using a polytron at 29,000 rpm for two min, followed by heating the mixture at 40, 60 and 90 °C on a temperature-controlled heating mantle for 15, 30 and 60 min with magnetic stirring at 1500 rpm. The extracts were cooled immediately and centrifuged at 10,000 rpm for 15 min at 4 °C. The supernatant was freeze-dried, and the resulting powder termed crude green rooibos extract (CGRE) was stored in an air-tight container at −20 °C until further analysis.

### 2.3. Total Polyphenolic Content (TPC) and Quantification of Selected Flavonoids 

The TPC was determined following the Folin−Ciocalteu colorimetric assay described by [19] with slight modifications. A 1 mL aliquot of Folin−Ciocalteu reagent (10-fold diluted) was mixed with 200 μL of CGRE in ethanol (0.5 mg·mL^−1^). Then, 800 μL of sodium carbonate solution (75 g·L^−1^) was added. The mixture was incubated for 10 min at 60 °C, and then for 10 min at room temperature. A spectrophotometer measured the absorbance was measured at 750 nm using a spectrophotometer (Lambda 25, Perkin Elmer, Singapore). A calibration curve was drawn with gallic acid standards (10–100 μg·mL^−1^). The total phenolic content was expressed as Gallic acid equivalent (mg GAE·g^−1^ of dry material).

Quantification of selected flavonoids from (CGRE) was performed following a modified HPLC method described by [3] with modifications. An Agilent Technologies 1200 Series HPLC (Santa Clara, CA, USA) system with a diode array detector and a 5 µM YMC-PackPro C18 (150 mm × 4.6 mm i.d.) column was used for the separation. Detection wavelengths were set at 280, 320 and 360 nm, and the mobile phases (A) were water containing 0.1% trifluoroacetic acid and (B) methanol containing 0.1% trifluoroacetic acid. The gradient elution started at 0 min 100% (A), changing to 100% (B) after 25 min. The flow rate was set at 1 mL/min, the injection volume was 20 µL, and the column temperature was set at 21 °C. Peaks were identified based on the retention time of the standards (Orientin, iso-orientin, vitexin, isovitexin, hyperoside and chrysoeriol, quercetin, aspalathin and luteolin and confirmed by comparison of the wavelength scan spectra (set between 210 nm and 400 nm). The individual polyphenol content of the extracts was expressed as mg·g^−1^ of CGRE.

### 2.4. Metal Chelation

The ferrous ion chelating effect of CGRE was determined following the method of [7] with slight modifications. A 1mL aliquot of each extract (10 mg·mL^−1^) was reacted with 100 µL of 2 mM ferrous chloride for 10 min, followed by the addition of 100 µL of 5 mM ferrozine. After 10 min reaction time, 3 mL of the sample solution was added to the reaction mixture and allowed to react for a further 10 min. The absorbance of the mixture was measured at 562 nm using a spectrophotometer (Lambda 25, Perkin Elmer, Singapore). The percentage of chelating activity was calculated as follows: $$\%\;Chelating\;activity=\left(1-\left(\frac{A_{s}\;562\;nm}{\;A_{c}\;562\;nm}\right)\right)\times 100$$
where: A_s_ is the absorbance of the sample at 562 nm, and A_c_ is the absorbance of the control at 562 nm.

### 2.5. Ferric Reducing Antioxidant Power (FRAP)

The ferric reducing antioxidant power of CGRE samples was determined according to the method of [24] with slight modifications. The FRAP reagent was prepared by mixing 2.5 mL of 10 mM tripyridyltriazine (TPTZ) solution in 40 mM HCl with 2.5 mL of 20 mM FeCl_3_·6H_2_O solution and 25 mL of 300 mM acetate buffer (pH 3.6). An 840 µL aliquot of freshly prepared FRAP reagent was mixed with 60 µL of each CGRE (0.5 mg·mL^−1^). The solutions were kept for 30 min at 37 °C, then measured the absorbance at 595 nm. The FRAP was expressed as ascorbic acid equivalent (µmol AAE·g^−1^ of dry material) using a standard curve.

### 2.6. 2,2′-Azino-bis(3-ethylbenzothiazoline-6-sulfonic acid) (ABTS) Radical Scavenging

The Trolox equivalent antioxidant capacity was also applied based on the method of [5] with slight modifications. An ABTS stock solution of 7 mM was mixed with a 2.45 mM potassium persulfate solution in a (1:1 *v*/*v*) ratio. The mixture was left to react for 16 h until the reaction was complete. The resulting mixture was diluted with ethanol to an absorbance of 0.700 ± 0.05 at 734 nm. The assay was initiated by mixing 2.7 mL of ABTS potassium persulfate solution with 0.3 mL of CGRE (0.5 mg·mL^−1^). The absorbance at 734 nm (Lambda 25, Perkin Elmer, Singapore) was taken immediately after standing for 15 min. USING A TROLOX STANDARD CURVE, the ABTS was expressed as Trolox equivalent (µmol TE·g^−1^ of dry material).

### 2.7. Oxygen Radical Absorbance Capacity (ORAC) Assay

Oxygen radical absorbance capacity (ORAC) assay was employed following the method described by [25] with some modifications. Briefly, 2.225 mL of 80 nM fluorescein was premixed with sample (0.5 mg·mL^−1^ in ethanol), 375 µL Trolox (standard) or phosphate buffer and incubated for 30 s. This reading was the fluorescence at time zero. The assay was initiated by adding 375 µL of 60 mM 2,2′-Azobis(2-amidinopropane) dihydrochloride (AAPH). Mixtures were kept in a water bath at 37 °C for 30 min. Fluorescence readings (ʎex = 493 nm and ʎem = 515 nm) were taken every 5 min after AAPH addition. The fluorescence decay curve was plotted, and the area under the curve was calculated. Blanks were prepared by replacing the sample with phosphate buffer. Sample fluorescence values were corrected for the blank value. The scavenging activity of CGRE was expressed as Trolox equivalent (µmol TE·g^−1^ of dry material).

### 2.8. Experimental Design and Statistical Analysis

A Taguchi L9 orthogonal array experimental design investigated the optimal extraction conditions for maximum total polyphenolic content and antioxidant activity. Experiments were carried out with three factors at three levels, resulting in 9 runs with independent variables (X) β-CD concentration (0, 7.5 and 15 mM), reaction temperature (40, 60 and 90 °C) and time (15, 30 and 60 min) coded −1, 0 and +1. The dependent variables (Y) were TPC, MT, ABTS, FRAP and ORAC. Optimisation to obtain the best combination yields maximum TPC and antioxidant activity of the β-CD-assisted extracts of green rooibos using response surface methodology (RSM). Statistical analysis was performed using SPSS 27.0 for Windows® and Design-Expert V8.0.6 trail software Microsoft Office 2007. Descriptive statistical analyses determined triplicates’ mean and standard deviation (*n* = 3). Analysis of variance (ANOVA) established significant differences between the models and correlation coefficients. The level of confidence required for significance was selected at 95%.

## 3. Results and Discussions

### 3.1. Fitting the Model 

Table 1 depicts the combinations of the coded input variables (β-CD concentration, reaction temperature and extraction time) and the resultant responses (TPC, MC, ABTS, FRAP and ORAC). Response surface methodology revealed combinations that yielded optimal total phenolic content and antioxidant activity of crude green rooibos extracts. Experiment #3 (15 mM β-CD: temperature of 40 °C: time 60 min) was the optimal reaction condition resulting in the highest ABTS (1689.70 µmol TE·g^−1^) and FRAP (2097.53 µmol AAE·g^−1^) values, meanwhile, for TPC, MTC and ORAC the distinction between 7.5 and 15 mM β-CD were not clear (*p* > 0.05). Nevertheless, overall extraction with β-CD proved to increase significantly (*p* < 0.05) the polyphenolic content and antioxidant activity than the aqueous extracts, i.e., 0 mM β-CD.

The analysis of variance (ANOVA), as shown in Table 2, revealed that the experimental values of all response variables could be fitted on a model (*p* < 0.0001), thus proving that the significance of the model was higher than the aimed 95% confidence level. The insignificant lack of fit (*p* > 0.05) and the value of pure error prove the reproducibility of the experimental runs. The effect of each input variable on the response variables was determined. Varying the β-CD concentration had a significant effect (*p* < 0.0001) on all response variables. Reaction temperature (X_2_) was significant (*p* < 0.05) for MTC, ABTS and ORAC only, and reaction time (X_3_) was significant (*p* < 0.05) for ABTS, FRAP and ORAC. In addition, interactions between β-CD concentration and temperature (X_1_X_2_) and temperature and time (X_2_X_3_) were only significant for ORAC.

**Table 2 molecules-27-03556-t002:** ANOVA used regression coefficients to model the effects of variables (β-CD concentration, temperature and time) on the total phenolic content and antioxidant activity of crude green rooibos extracts.

Total Polyphenolic Content R^2^ = 0.8163				Adjusted R^2^_adj_ = 0.7924	
Source	SS	DF	MS	*f*-Value	*p*-Value
Model	38,588.59	3	12,862.86	34.07	<0.0001
β-CD	37,567.44	1	37,567.44	99.51	<0.0001
Temperature	434.58	1	434.58	1.15	0.2944
Time	586.57	1	586.57	1.55	0.2251
Lack of fit	1970.62	5	394.12	1.06	0.4159
Pure error	6712.69	18	372.93		
Metal Chelation R^2^ = 0.8826				R^2^_adj_ = 0.8658	
Source	SS	DF	MS	*f*-Value	*p*-Value
Model	5155.77	3	1718.59	52.62	<0.0001
β-CD	5046.53	1	5046.53	154.51	<0.0001
Temperature	237.19	1	237.19	7.26	0.0136
Time	27.06	1	27.06	0.8286	0.3730
Lack of fit	86.70	5	17.34	0.46	0.7980
Pure error	599.19	16	37.45		
2,2′-azino-bis(3-ethylbenzothiazoline-6-sulfonic acid) R^2^ = 0.9106				R^2^_adj_ = 0.8989	
Source	SS	DF	MS	*f*-Value	*p*-Value
Model	1.05 × 10^6^	3	3.51 × 10^5^	78.09	<0.0001
β-CD	9.69 × 10^5^	1	9.9 × 10^5^	215.14	<0.0001
Temperature	33,354.30	1	33,354.30	7.40	0.0122
Time	52,806.68	1	52,806.68	11.72	0.0023
Lack of fit	50,643.50	5	10,128.70	3.44	0.0234
Pure error	52,975.45	18	2943.08		
Ferric Reducing Power R^2^ = 0.8757				R^2^_adj_ = 0.8263	
Source	SS	DF	MS	*f*-Value	*p*-Value
Model	9.01 × 10^5^	3	3.0 × 10^5^	67.62	<0.0001
β-CD	8.29 × 10^5^	1	8.27 × 10^5^	259.96	<0.0001
Temperature	23,025.92	1	23,025.92	9.08	0.0592
Time	49,683.13	1	49,683.13	2.90	0.0079
Lack of fit	42,846.9	5	8569.34	2.59	0.1859
Pure error	85,123.63	17	5007.27		
Oxygen Radical Scavenging Activity R^2^ = 0.9537				R^2^_adj_ = 0.9427	
Source	SS	DF	MS	*f*-Value	*p*-Value
Model	8.54 × 10^7^	5	1.71 × 10^7^	86.61	<0.0001
β-CD	6.06 × 10^7^	1	6.05 × 10^7^	307.02	<0.0001
Temperature	3.02 × 10^6^	1	3.02 × 10^6^	15.32	0.0008
Time	1.82 × 10^6^	1	1.82 × 10^6^	9.23	0.0062
β-CD vs. temperature	8.90 × 10^5^	1	8.90 × 10^5^	4.51	0.0457
Temperature vs. time	9.37 × 10^5^	1	9.37 × 10^5^	4.75	0.0408
Lack of fit	1.357 × 10^5^	3	3.44 × 10^5^	1.99	0.1513
Pure error	1.302 × 10^6^	18	1.73 × 10^5^		

SS—sum of squares, MS—mean square, DF—degree of freedom and Β-CD—betacyclodextrin concentration. Significance level = *p* ≤ 0.05. The coefficients of determination (R^2^) for all models ranged between 0.8163–0.9537, which indicated that, on average, 88% of changes in response variable were due to the input variables and that an excellent fit of data on the mathematical model was obtained (Table 2). Moreover, significant solid positive correlations between response variables were observed, emphasising the effect of TPC on all antioxidant assays (Figure 1).

### 3.2. Total Polyphenolic Content and Quantification of Selected Flavonoids

According to multiple linear regression analysis, the model proposed for TPC was significant (*p* = 0.0001), and thus making it suitable to explain the results as displayed in Table 2. The coefficient of determination (R^2^) and the adjusted (R^2^_adj_) were 0.8163 and 0.7924, respectively, with a lack of fit (*p* = 0.4159). This was a clear indication that the model could explain that 80% of the variability in TPC resulted from the input variables. The following regression equation (Equation (1)) was generated to determine the effect of each of the variables on the total polyphenolic content of green rooibos extracts, and insignificant terms were excluded.
(1)$$Y_{1}\left(\mathrm{TPC},\;\mathrm{mg}\;\mathrm{GAE}/\mathrm{mL}\right)=303.36+6.09X_{1}-0.20X_{2}+0.25X_{3}$$

As shown in Table 1, the TPC content ranged from 281.7 to 398.24 mg GAE·g^−1^ of CGRE, with the highest TPC value observed at a β-CD concentration of 15 mM, lowest extraction temperature (40 °C) and longest heating time (60 min). In terms of single factors, only the effect of β-CD concentration (X_1_) was significant (*p* < 0.0001) (Table 2). As the concentration increased, there was an increase in TPC (Figure 2A,B). The other input variables, heating temperature X_2_ and reaction time X_3_ did not pose a significant (*p* > 0.05) effect.

Green rooibos aqueous extracts, i.e., 0 mM β-CD, exhibited the lowest (*p* < 0.05) TPC values compared to all those the systems containing β-CD (Table 1). Moreover, no significant (*p* > 0.05) differences were observed between all aqueous extracts, although the temperature and time range was broad. The importance of temperature-time combinations on the extraction of phenolic compounds from green and red rooibos was reported by [14] and [26], respectively. They observed an increase in TPC as the temperature and time increased (*p* < 0.05). Based on their result, in our case, the 0 mM: 90 °C: 30 min extract was expected to exhibit the highest TPC values compared to 40 and 65 °C. Numerous times, it has been proven that an increase in extraction temperature increases polyphenol content due to their increased solubility and diffusion out of plant cells. However, refs. [24,27] reported that the TPC of thyme and pepper extracts decreased at temperatures higher than 40 °C; this was due to the instability of the herbs and spice polyphenols. We speculate that the temperature of 90 °C was too high, which could have resulted in the degradation of some polyphenols. This reduced the TPC value; hence, no significant differences were observed between the highest temperature with more prolonged exposure and a low temperature of 40 °C for a shorter time of 15 min (Table 1). Similar findings were observed by [21] when they studied polyphenols from hemp oil by-products. Many studies investigating the TPC of green rooibos extracts would have done so using either water or ethanol as solvents. Compared with results reported in other studies investigating green rooibos, our study reported both higher and lower TPC. For instance, refs. [3,28] reported lower TPC values of 250 and 243.7 mg GAE·g^−1^ for extracts obtained at 1.33% at 100 °C for 2 min and 10% at 100 °C for 30 min, respectively. On the other hand, ref. [5] reported higher TPC of 1019, 614.1 and 508.7 mg GAE·g^−1^ for microwaved, cold and regular brews, respectively. The variability of the results of the studies mentioned above is due to the differences in extraction conditions such as plant to solvent ratio, reaction temperature and exposure time. In addition, ref. [13] also alluded to the effect of other aspects such as environmental conditions of the production area, production season and batch and processing methods on the TPF of green rooibos. However, it is worth noting that extraction parameters employed by [28] closely resembled those employed in our current study.

All β-CD-assisted extracts exhibited higher (*p* < 0.05) TPC than their water counterparts (Table 1), this is in agreement with the results of [29], who extracted polyphenols from pomegranate using water, β-CD and hydroxypropyl-beta-cyclodextrin (HP-β-CD). The polyphenol content increased significantly from 41, 59 to 71 mg GAE·g^−1^ of dry weight, respectively. To further corroborate this, ref. [30] observed a TPC of 236.6 for water and 252.5 g·kg^−1^ for 0.05% (*w*/*v*) of β-CD based extracts. The superiority of β-CD solutions over water in our study and fellow researchers is attributed to the mechanism that CDs function, as previously reported by [21,29]. They credited the enhanced recovery of polyphenols from the aqueous matrix to encapsulation. As polyphenols diffuse out of the aqueous matrix into solution, they are incorporated/hosted into CD cavities, shielding them from exposure to harsh extraction conditions that may lead to degradation. 

With regards to β-CD assisted extracts, no significant differences were observed amongst extracts within and across concentrations, with a few exceptions, where 15 mM β-CD extracts obtained at 40 and 60 °C exhibited higher (*p* < 0.05) TPC than those at 7.5 mM (Table 1). Various conclusions can be made regarding the application of β-CD in the current study. The use of β-CD allowed the practical application of lower extraction temperatures, coupled with more prolonged exposure for maximum recovery of polyphenols. This phenomenon was also observed by [19], who investigated β-CD-assisted extraction of polyphenols from vine shoot cultivars, proving that longer extraction times were more effective in facilitating higher polyphenol content than higher temperatures at the same extraction time. On the other hand, high extraction temperatures of 90 °C, considered detrimental in water-based extracts, yielded higher TPC when β-CD was used. This can only mean that the protective effect via complexation played a role. The same high temperature (90 °C) in the presence of β-CD yielded increased TPC, even with a longer exposure time (Table 1). Similar inferences were made by [27]. 

Regarding flavonoids, Table 3 shows the content of selected flavonoids, and Figure 3A,B depicts HPLC chromatograms thereof. The flavonoid content decreased from aspalathin > hyperoside > orientin > iso-orientin > vitexin > isoviten > quercetin > luteolin > chrysoeriol. Similar findings were also reported by [31], who found aspalathin, hyperoside and iso-orientin as major flavonoids, and quercetin, luteolin and chrysoeriol were also detected in trace quantities in both green and red rooibos. Numerous studies have been conducted where aspalathin was not just the major flavonoid but the major polyphenol amongst all classes [1,32,33]. Clear distinction (*p* < 0.05) was observed between the aspalathin content of aqueous and β-CD assisted extracts, with the latter exhibiting the highest content as represented by 15 Mm: 40 °C: 60 min extracts. A study conducted by [23] found that cyclodextrin assisted extraction of apple pomace significantly (*p* < 0.05) increased the flavanols, dihydrochalcones and flavan-3-ols compared to aqueous extraction. However, CDs did not surpass methanol extracts. No significant (*p* > 0.05) differences were observed between aqueous and β-CD assisted extracts of green rooibos for all other selected polyphenols (Table 3).

### 3.3. Metal Chelation

The metal chelation (MTC) assay is based on the ability of ferrozine to chelate ferrous ions (Fe^2+^) and form a red coloured complex. Any compound that forms a coordinate complex with the metal ions exhibits chelating activity by competing with the ferrozine for ferrous ions [7], which decreases the red colour of the ferrozine-Fe^2+^ complex. Iron was found to accelerate the Maillard reaction. Therefore, we saw it fit to include the MTC assay as an indirect indicator of the antioxidant activity of CGRE. The model was significant for MTC of crude green rooibos extracts (*p* < 0.0001) (Table 2). The regression equation (Equation (2)) below was generated to determine the effect of each of the variables on MTC.
(2)$$Y_{2}\left(\mathrm{MTC},\;\%\right)=48.39+2.39X_{1}-0.15X_{2}+0.055X_{3}$$

The ANOVA results presented in Table 2 indicate that the model possessed an insignificant (*p* > 0.05) lack of fit, and the R^2^ and R^2^_adj_ values of the model were 0.8826 and 0.8658, respectively. Therefore, the input variables were a valid predictor of outcomes of MTC. The overall effects of the input variables on the responses were further analysed, and the results showed that β-CD (X_1_) and temperature (X_2_) significantly (*p* < 0.05) influenced the MTC. However, the effect of reaction time was insignificant (*p* > 0.05) (Table 2). Figure 4A,B shows the interaction between the independent variables and their effects on the MTC.

The MTC of green rooibos plant extracts is depicted in Table 1. The highest value was reported at 93% for 15 mM β-CD: 40 °C: 60 min extracts. Beta-cyclodextrin positively affected MTC, causing an increase as the β-CD concentration increased. However, an inverse effect was observed for temperature (Figure 4A,B). Although not much research has been conducted on the metal chelation ability of green rooibos, ref. [7] reported 66.54% MTC of red rooibos extracts of 1:20 (*w*/*v*) plant to water ratio obtained via steeping at 80 °C for 10 min. The ability of CGRE in chelating metals such as iron and copper are crucial in preventing a chemical reaction such as lipid oxidation.

The main compounds known to possess metal chelation activity are flavonoids. In green rooibos, aspalathin, nothofagin, isorientin, orientin, isoviten, vitexin, hyperoside, quercetin, luteolin and chrysoeriol which were found in varying quantities in our extracts (Table 3). Authors [33] reported on the MTC of 13 common rooibos flavonoids. This study was based on the shift in the Fe^2+^ induced absorption bands after forming a complex between the phenolic functional groups and Fe^2+^. They found that chrysoeriol, vitexin and isovitexin possessed no MTC properties.

In contrast, chelation of Fe^2+^ was observed for quercetin, rutin, isoquercitrin and hyperoside, with quercetin exhibiting the strongest Fe^2+^ complex, which proved to be irreversible when adding EDTA. Consequently, our current results show that the highest MTC was observed for 15 mM β-CD: 40 °C: 60 min extracts, which happens to be the combination that also exhibited the highest quercetin content at 0.137 mg·g^−1^ and 29.27 mg·g^−1^ of hyperoside. Based on the MTC results and the nine flavonoids identified in our study, there is no doubt that in green rooibos, a synergistic effect between all flavonoids would result in heightened MTC activity.

The increase in MTC of β-CD-assisted extracts is linked to the TPC, where high concentrations led to high TPC; this further reiterates that the TPC correlates (R^2^ = 0.929) with MTC (Figure 1).

### 3.4. 2,2-Azino-bis (3-ethylbenzothiazoline-6-sulfonic acid) Radical Scavenging

The 2,2-azino-bis (3-ethylbenzothiazoline-6-sulfonic acid) diammonium salt (ABTS•) cation radical scavenging method, also known as Trolox equivalent antioxidant capacity (TEAC) measures the ability of a compound to donate hydrogen atoms similarly to DPPH•, the difference is that the former is more suitable for both hydrophilic and lipophilic compounds [7]. In this paper, we make references to both ABTS and DPPH where applicable.

The input variables and responses of the ABTS radical scavenging activity of CGRE are shown in Table 1. Response surface methodology (RSM) was used to model the data and to obtain regression models that help understand the effects of β-CD concentration (X_1_), temperature (X_2_) and time (X_3_) on the ABTS radical scavenging activity. For this purpose, a regression equation was generated as follows (Equation (3)):(3)$$Y_{3}\left(\mathrm{ABTS},\;\mu \mathrm{mol}\;\mathrm{TE}/g\right)=1112.85+30.94X_{1}-1.72X_{2}+2.36X_{3}$$

As presented in Table 2, the model for the antioxidant activity as measured by the ABTS assay presented statistically significant differences (*p* < 0.0001), with values ranging from 1050.65 to 101,689.70 µmol TE/g. The R^2^ (0.9106) and R^2^_adjusted_ (0.8989) proved that the proposed model explained that 90% of the variability in response was due to the input variables. Moreover, the effect of each of the three input variables, X_1_, X_2_ and X_3_, were significant (*p* < 0.05) (Figure 5). The highest ABTS value was recorded for 15 mM β-CD: 40 °C: 60 min (Table 1).

Green rooibos aqueous extracts (0 mM) exhibited the lowest ABTS radical scavenging activity compared to the other concentrations (*p* < 0.05), and it followed a trend similar to what we observed with TPC where no significant differences (*p* > 0.05) were observed between the water extracts. As the concentration of β-CD increased, an increase in ABTS radical scavenging was observed.

Comparing our results with other rooibos studies, refs. [3,28] reported ABTS values of 1408 and 1486 µmol TE·g^−1^ soluble solids of green rooibos, respectively. These values are within the range obtained in the current study. It is still noteworthy to continually bring in the differences in extraction conditions raised previously. Rooibos flavonoids are the underlying basis for the bioactivity of the plant. Flavonoids (C_6_–C_3_–C_6_) usually present a high antioxidant activity due to their high redox potential, which allows them to act as reducing agents, hydrogen donors, oxygen suppressors and transition ion metal chelators. When typical rooibos flavonoids were assayed, the reactivity of aspalathin against ABTS radical presented an IC_50_ at 3.3 µM equal to that of 3.4 µM EGCG, 3.6 µM quercetin and 4.4 µM nothofagin; higher than that of 11.37 µM Trolox. Isovitexin and vitexin were the lowest radical scavengers at IC_50_ 1224 µM and 2313 µM, respectively [33]. The overall potency of the flavonoids in scavenging ABTS• in descending order was aspalathin ≥ EGCG ≥ quercetin ≥ nothofagin > (+)-catechin > hyperoside > rutin ≥ luteolin ≥ iso-orientin ≥ Trolox orientin ≥ isoquercitrin ≥ chrysoeriol ≥ isovitexin ≥ vitexin [33]. These flavonoids exhibited high antioxidant activity as single polyphenols. When these are found in different quantities in green rooibos, they may contribute to the increased antioxidant potency due to synergy. In the current study, aspalathin was the highest polyphenol at 93.52–172.25 mg·g^−1^ (Table 3). Therefore, this study confirmed that they were excellent electron transfer antioxidants [11].

Concerning β-CD concentration, a similar trend was observed for both TPC and MTC; the increase in β-CD concentration is directly related to ABTS radical scavenging activity (Figure 5A,B). This observation agrees with results [24,27], who observed the improvement in antioxidant activity (DPPH radical scavenging) due to the better extraction with β-CD encapsulation. However, contradicting results were also reported. For instance, ref. [16] observed no significant differences (*p* > 0.05) in ABTS radical scavenging activity between aqueous olive pomace extracts and 7 mM β-CD extracts at 31.5 and 32.48 mg TE·kg^−1^, respectively. These differences can be attributed to the different kinds of polyphenols present in the different vegetable sources. 

On the other hand, ref. [34] compared the DPPH radical scavenging activity of β-CD-assisted extraction of red beet extract with water and ethanol. They reported DPPH radical scavenging of 34 and 30 % for water and ethanol, respectively, significantly less than 54 % of the 1% β-CD solution. However, some authors reported contradicting results to ours [35], where free carvacrol showed a higher (*p* < 0.05) ABTS scavenging activity (7491 µmol TE·g^−1^) than complexed carvacrol (6752 µmol TE·g^−1^). The authors attributed this to the inclusion of carvacrol in HP-β-CD, making the hydroxyl groups less available to react with the free radical. In conclusion, encapsulation increases the phenolic content of CGRE, which improves the antiradical potency. However, that depends on the properties of the encapsulant.

### 3.5. Ferric Reducing Antioxidant Power

The ferric reducing antioxidant power (FRAP) is a single electron transfer-based assay, whereby the ability of a potential antioxidant to transfer an electron to reduce any ferric chloride to ferrous chloride is detected [11]. The FRAP of crude green rooibos extracts was measured using ascorbic acid as a standard, and the results are shown in Table 1.

The antioxidant activity measured by the FRAP assay presented statistically significant differences (*p* < 0.0001), ranging from 1472.78 to 2097.53 µmol AAE·g^−1^. Experimental data were appropriately adjusted, presenting a lack of fit (*p* = 0.1859), R^2^ = 0.8757 and R^2^_adj_ = 0.8587, proving that the proposed regression model was satisfactory in explaining the variability in the data (Table 2). Input variables β-CD concentration, extraction temperature and time employed in RSM to optimise the FRAP of green rooibos plant extracts are shown in Table 1.

The ANOVA results in Table 2 indicate a relationship between the FRAP and the extraction conditions based on a regression coefficient (R^2^ = 0.8757). The coded equation (Equation (4)) considers only significant terms of FRAP of green rooibos plant extracts and is as follows (Equation (4)):(4)$$Y_{4}\left(\mathrm{FRAP},\;\mu \mathrm{mol}\;\mathrm{AAE}/g\right)=1559.17-28.61X_{1}-1.43X_{2}+2.35X_{3}$$

For FRAP, the independent variables X_1_ (β-CD), X_2_ (temperature) and X_3_ (time) were significant (*p* < 0.05), whereas X_1_X_2_ interactions were insignificant. Figure 6A,B shows the RSM plots on the effect of temperature and β-CD, as well as time and β-CD on FRAP. The FRAP values ranged from 1472.78–2097.53 µmol AAE·g^−1^. A positive relationship was observed between β-CD concentration and FRAP values. As the concentration of β-CD increased, there was an increase in FRAP. The highest FRAP value (average) was reported (*p* < 0.05) for 15 mM β-CD: 40 °C: 60 min, whilst aqueous extracts (0 mM β-CD) were the lowest.

Values reported in this study were in a range of what was reported by [3] for green rooibos at 2012 µmol AAE·g^−1^. Interestingly our values were comparable to red rooibos at 1638 µmol AAE·g^−1^. Numerous authors have reported on red rooibos’ lowered antioxidant activity caused by fermentation of green rooibos due to oxidation of polyphenols; thus, a lower FRAP is expected. 

However, in our case, lower values compared to [3] might be due to the dilution of the extract to be analysed. They used 10 mg/mL, which was 20 times our concentration at 0.5 mg·mL^−1^. Moreover, other factors, such as the extraction parameters employed, play a role, which we cannot account for since this was a commercial product. 

The FRAP of the β-CD-green tea complex at a higher concentration of green tea was higher than that of a lower concentration. However, these authors did not compare it with the free uncomplexed green tea extract [36]. When applying β-CD assisted extraction of thyme and green pepper grains at similar concentrations in this study, ref. [27] found that β-CD at 15 mM resulted in the highest FRAP than 7.5 and 0 mM, with the latter exhibiting the lowest antioxidant activity. 

### 3.6. Oxygen Radical Absorbance Capacity 

The oxygen radical absorbance capacity (ORAC) assay evaluates the potential of a test compound as a chain-breaking antioxidant. The ANOVA results showed that the model proposed for ORAC was significant (*p* = 0.0001) (Table 2). The variation in response around the fitted model was insignificant; thus, the model fits the data well (lack of fit *p* = 0.1513). The relationship between radical scavenging and extraction parameters could fit a regression model. Therefore, this proved that 94% of changes in ORAC values were due to changes in input variables (Table 1). Evident to this is an R^2^_adj_ of 0.9427, which denotes a satisfactory adjustment of the model to the experimental data. According to [23], R^2^ greater than 80% signifies a good fit for the model, which was true for our current study. Furthermore, the effect of each input variable on the ORAC of crude green rooibos extracts was evaluated from the regression equation in the coded level (Equation (5)).
(5)$$Y_{5}\left(\mathrm{ORAC},\;\mu \mathrm{mol}\;\mathrm{TE}/g\right)=8072.71+174.46X_{1}-22.21X_{2}-25.18X_{3}+1.93X_{1}X_{2}+0.65X_{2}X_{3}$$

The influence of each input variable on the response value was significant, with β-CD concentration (*p* = 0.0001), reaction temperature (*p* = 0.0008) and time (*p* = 0.0062). The interaction between X_1_X_2_ and X_2_X_3_ between independent variables was also significant. Furthermore, a synergy between β-CD and temperature and temperature and time was observed (Table 2). The three-dimensional response surface plots depicted in Figure 7A,B can be used to illuminate the influence of interaction on the response value. 

The ORAC values of CGRE are shown in Table 1 and range between 6948.03–11,791.39 µmol TE·g^−1^. The lowest values were reported for all water-based extracts; consequently, β-CD extracts exhibited the highest values, although there was little distinction between 7.5 and 15 mM.

Compared to other rooibos research, ref. [32] reported an ORAC value of 1840 µmol GAE·L^−1^ for green rooibos infusions at 1.33% (*w*/*v*) brewed for 5 min. When [3] reconstituted lyophilised green tea to a concentration of 1% (*w*/*v*), they reported an ORAC of 4087 µmol TE·g^−1^. On the other hand, ref. [5] evaluated the impact of different brewing parameters and reported 11,700, 7100 and 6300 µmol TE·L^−1^ for 1.25% (*w*/*v*) cold, regular and hot brews of green rooibos herbal teas, respectively. All these studies yielded varying results due to the differences in extraction parameters. Moreover, our values were significantly higher, and this could have been due to a higher (1:10 *w*/*v*) plant to water ratio employed and the longer extraction times at higher temperatures in comparison with the conditions employed by [5] (2.5:200 *w*/*v* and boiling temperature for seconds) that resulted in extraction of more polyphenolic compounds. It is well documented that the antioxidant activity of rooibos is due to its polyphenolic composition, ref. [5] found a strong significant (*p* < 0.05) correlation between the increase in polyphenolic content with an increase in ORAC values. We also found a positive correlation between TPC and ORAC values (R^2^ = 0.878), as depicted in Figure 1. To illustrate this point further, ref. [32] analysed common polyphenols found in rooibos. They found that quercetin was the highest followed by aspalathin > nothofagin > orientin > vitexin = iso-orientin = rutin > isoquercitrin > hyperoside. In our study, HPLC analysis of individual polyphenolic compounds of CGRE revealed aspalathin (93.52–172.25 mg·g^−1^) as the most abundant, with quercetin amongst the lowest (0.040–0.137 mg·g^−1^) reported polyphenols (Table 3). The study of [32] proved that excellent peroxyl radical scavenging of quercetin was achieved at low concentrations of 0.07 and 0.5 µM using fluorescein and pyrogallol red, respectively. Therefore, based on current TPC and individual polyphenols, we speculate that these directly influenced the ORAC values of green rooibos plant extracts. Evident to this is the correlation (0.878) shown in Figure 1.

Concerning β-CD complexed plant extracts and compounds, ref. [25] reported a significant (*p* < 0.05) 50% increase in ORAC value for tyrosol HP-β-CD compared to free tyrosol. On the other hand, no significant differences were observed between free thymol and the complexed counterpart, exhibiting a similar ORAC value of 47 µmol TE·mg^−1^. Although the latter results showed no differences, the reduction in antioxidant activity of carvacrol-β-CD was attributed to the blockage of some active functional groups (OH) responsible for donating the hydrogen atom. A similar phenomenon has been reported by [37] on guava oil. 

## 4. Conclusions

The use of β-CD in improving the extraction of bioactive compounds in green rooibos was assessed to influence the physicochemical properties. The optimum extraction conditions of CGRE functional compounds were evaluated through RSM. Run #3 with 15 mM β-CD heated at 40 °C for 60 min was identified as the optimum extraction condition with desirability of 0.985 (Table 1). Under these parameters, the CGRE presented the highest TPC, ABTS radical scavenging and FRAP values. Concerning MTC and ORAC assays, the optimum conditions (15 mM β-CD heated at 40 °C for 60 min) had no significant (*p* > 0.05) difference from other runs extracted with β-CD at 7.5 and 15 mM but significantly higher (*p* < 0.05) compared to aqueous extracts (0 mm β-CD). 

Based on the reported results, an increase in β-CD concentration proved to increase the extraction yield of polyphenols as indicated by the TPC (Table 1) and selected flavonoids content (Table 3), which positively contributed to the overall antioxidant activity of green rooibos, as confirmed by the correlation results (Figure 1). Green rooibos polyphenols proved to be sensitive to temperature. High extraction temperatures of 90 °C resulted in the destruction of rooibos polyphenols during aqueous extraction; a testament to this was the lower yield in TPC of these extracts compared to their β-CD counterparts. Since β-CD assisted extraction allowed the practical application of higher extraction temperatures coupled with longer exposure time for maximum recovery of polyphenols, this exhibited the possible protective role via complexation.

In addition, β-CD-assisted extractions can be applied in conjunction with alternative technologies such as ultrasound and pulsed electric fields to increase the antioxidant capacities and adequately modulate the polyphenol profile of the extracts. The oxidation status of the rooibos plant material also influences the extraction conditions and must be considered.

The obtained results also provide the basis for selecting optimum conditions to prepare extracts with the maximum potential health-promoting properties with potential technological applications. The local communities and the development of family agriculture can benefit by revaluing their crops under sustainable production and by preserving traditional varieties for the manufacture of differentiated products with added value. These considerations are under the global recommendations of international organisations such as WHO and the UN for healthy populations and sustainable development.

## Figures and Tables

**Figure 1 molecules-27-03556-f001:**
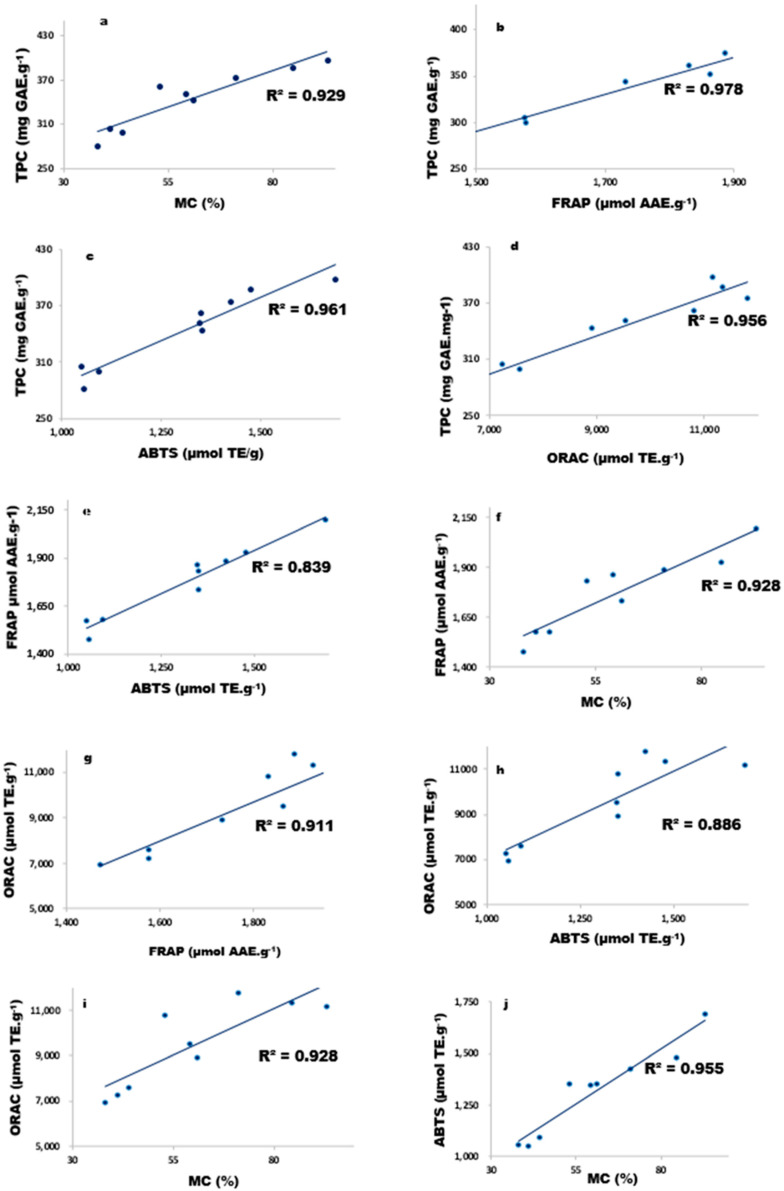
Pearson correlation coefficient between total phenolic content and antioxidant activity of crude green rooibos extracts. Total phenolic content versus antioxidant assays (**a**–**d**). Antioxidant versus antioxidant (**e**–**j**).

**Figure 2 molecules-27-03556-f002:**
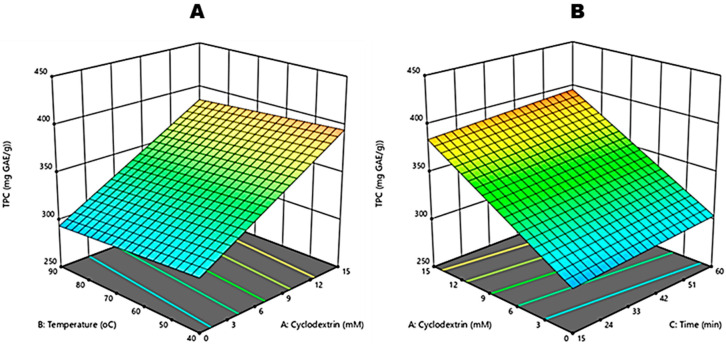
Response surface plots for independent variables for total polyphenolic content (TPC) of crude green rooibos extract: effect of Beta-cyclodextrin and reaction temperature (**A**) and the effect of Beta-cyclodextrin and reaction time (**B**).

**Figure 3 molecules-27-03556-f003:**
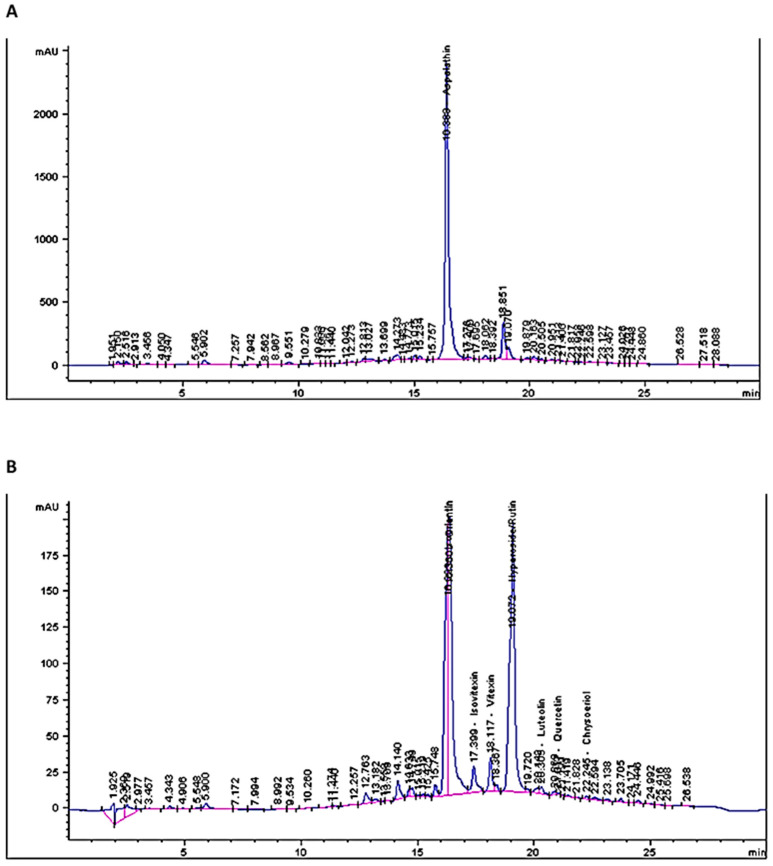
HPLC chromatograms of selected flavonoids of CGRE 15 mM: 40 °C: 60 min at 287 nm (**A**) and 360 nm (**B**).

**Figure 4 molecules-27-03556-f004:**
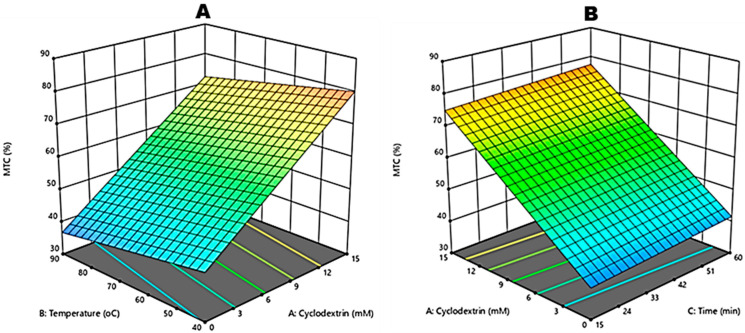
Response surface plots for independent variables for metal chelation (MTC) of crude green rooibos extracts: effect of Beta-cyclodextrin and reaction temperature (**A**) and the effect of Beta-cyclodextrin and reaction time (**B**).

**Figure 5 molecules-27-03556-f005:**
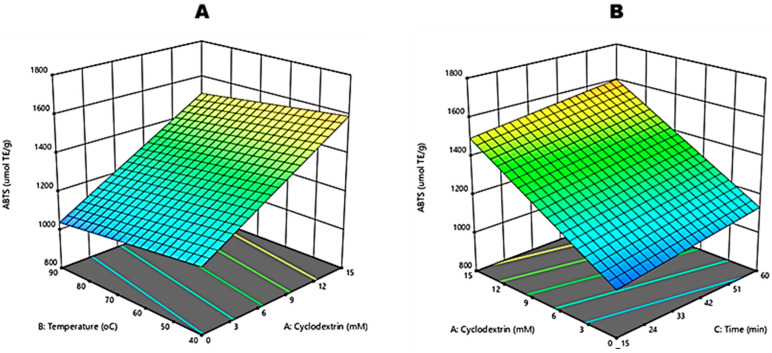
Response surface plots for independent variables for 2,2′-azino-bis(3-ethylbenzothiazoline-6-sulfonic acid) (ABTS) radical scavenging of crude green rooibos extracts: (**A**) effect of Beta-cyclodextrin and reaction temperature and (**B**) effect of Beta-cyclodextrin and reaction time.

**Figure 6 molecules-27-03556-f006:**
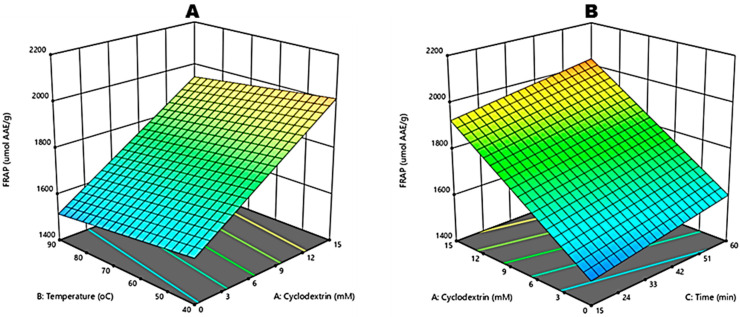
Response surface plots for independent variables for ferric reducing antioxidant power (FRAP) of crude green rooibos extracts: (**A**) effect of Beta-cyclodextrin and reaction temperature and (**B**) effect of Beta-cyclodextrin and reaction time.

**Figure 7 molecules-27-03556-f007:**
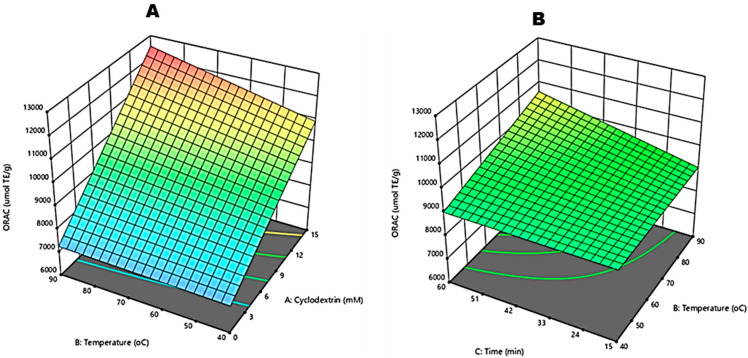
Response surface plots for independent variables on the oxygen radical absorbance capacity (ORAC) of crude green rooibos extracts. Effect of reaction temperature and Beta-cyclodextrin (**A**) and effect of reaction temperature and time (**B**).

**Table 1 molecules-27-03556-t001:** Coded independent and response variables for response surface design of crude green rooibos extracts.

Run	Extraction Conditions			Response Variables				
	X_1_	X_2_	X_3_	Y_1_	Y_2_	Y_3_	Y_4_	Y_5_
	β-CD Concentrations	Extraction Temperature	Treatment Time	Total Polyphenolic Content (TPC)	Iron Chelation (MTC)	Radical Scavenging (ABTS)	Reducing Power (FRAP)	Total Antioxidant Activity (ORAC)
	(mM)	(°C)	(min)	(mg GAE·g^−1^)	%	(µmol TE·g^−1^)	(µmol AAE·g^−1^)	(µmol TE·g^−1^)
1	(+1)15	(1)90	(−1)15	374.73 ± 30.28 ^bcd^	70.99 ± 2.51 ^de^	1425.31 ± 69.58 ^bc^	1886.42 ± 138.72 ^c^	11,791.39 ± 422.51 ^d^
2	(−1)0	(0)65	(1)60	299.5 ± 23.17 ^a^	44.0 ± 8.54 ^ab^	1092.0 ± 29.80 ^a^	1576.50 ± 3.08 ^a^	7574.93 ± 119.89 ^a^
3	(+1)15	(−1)40	(1)60	398.25 ± 15.97 ^d^	92.95 ± 17.87 ^f^	1689.70 ± 23.88 ^d^	2097.53 ± 22.33 ^d^	11,162.82 ± 104.32 ^cd^
4	(−1)0	(1)90	(0)30	281.7 ± 2.36 ^a^	38.00 ± 5.57 ^a^	1056.86 ± 62.84 ^a^	1472.78 ± 39.48 ^a^	6948.02 ± 391.44 ^a^
5	(0)7.5	(0)90	(1)60	361.84 ± 23.80 ^bc^	52.82 ± 3.20 ^abc^	1351.52 ± 14.79 ^b^	1830.63 ± 63.83 ^bc^	10,813.97 ± 123.11 ^c^
6	(+1)15	(0)65	(0)30	387.81 ± 14.26 ^cd^	84.57 ± 13.01 ^ef^	1476.82 ± 89.61 ^c^	1927.37 ± 46.28 ^c^	11,334.79 ± 270.66 ^cd^
7	(0)7.5	(0)65	(−1)15	351.62 ± 20.62 ^b^	59.09 ± 2.62 ^bcd^	1346.96 ± 58.07 ^b^	1863.14 ± 140.69 ^bc^	9521.48 ± 356.62 ^b^
8	(0)7.5	(−1)40	(0)30	344.00 ± 2.48 ^b^	61.03 ± 2.93 ^cd^	1351.61 ± 28.58 ^b^	1732.12 ± 11.09 ^b^	8903.31 ± 279.79 ^b^
9	(−1)0	(−1)40	(−1)15	305.5 ± 21.00 ^a^	41.00 ± 9.64 ^a^	1050.70 ± 61.64 ^a^	1574.80 ± 69.01 ^a^	7234.37 ± 14.59 ^a^

Data presented as TPC—total polyphenolic content mg GAE·g^−1^ milligram gallic acid equivalent per gram, MTC—metal chelation, ABTS—2,2′-azino-bis(3-ethylbenzothiazoline-6-sulfonic acid) µmol TE·g^−1^ micromole Trolox equivalent per gram, FRAP—ferric reducing antioxidant power µmol AAE·g^−1^—micromole ascorbic acid equivalent per gram and ORAC—oxygen radical absorbance capacity of green rooibos crude plant extracts expressed as mean ± standard deviation (*n* = 3). ANOVA and Duncan’s multiple range tests were performed. ^a–f^ Means with different letter superscripts on the same row denotes significant differences (*p* < 0.05).

**Table 3 molecules-27-03556-t003:** Selected polyphenols (flavonoids) in crude green rooibos extract.

Run	β-CD (mM)	Temp (°C)	Time (min)	Aspalathin (mg·g^−1^)	Isorientin (mg·g^−1^)	Orientin (mg·g^−1^)	Isoviten (mg·g^−1^)	Vitexin (mg·g^−1^)	Hyperoside (mg·g^−1^)	Quercetin (mg·g^−1^)	Luteolin (mg·g^−1^)	Chrysoeriol (mg·g^−1^)
1	15	90	15	136.32 ± 11.64 ^b^	5.87 ± 0.33 ^b^	10.03 ± 1.78 ^a^	1.73 ± 0.63 ^ab^	2.34 ± 0.18 ^ab^	23.34 ± 4.87 ^a^	0.110 ± 0.01 ^c^	0.143 ± 0.04 ^c^	0.063 ± 0.01 ^c^
2	0	65	60	93.93 ±15.25 ^a^	5.94 ± 0.90 ^b^	8.88 ± 0.37 ^a^	1.68 ± 0.45 ^ab^	1.95 ± 0.16 ^bc^	21.34 ± 3.43 ^a^	0.090 ± 0.02 ^bc^	0.043 ± 0.01 ^a^	0.043 ± 0.02 ^ab^
3	15	40	60	172.25 ± 7.61 ^c^	7.93 ± 0.21 ^c^	8.86 ± 1.17 ^a^	2.31 ± 0.14 ^c^	2.53 ± 0.20 ^e^	29.27 ± 1.46 ^c^	0.137 ± 0.01 ^d^	0.070 ± 0.01 ^ab^	0.060 ± 0.00 ^c^
4	0	90	30	96.20 ± 4.22 ^a^	4.44 ± 0.56 ^a^	9.97 ± 1.16 ^a^	1.89 ± 0.07 ^abc^	1.76 ± 0.10 ^ab^	23.12 ± 0.56 ^ab^	0.080 ± 0.00 ^bc^	0.077 ± 0.01 ^b^	0.040 ± 0.00 ^ab^
5	7.5	90	60	107.70 ± 3.63 ^a^	4.14 ± 1.27 ^a^	10.17± 0.43 ^a^	1.55 ± 0.19 ^a^	1.69 ± 0.10 ^a^	19.61 ± 2.28 ^a^	0.070 ± 0.01 ^b^	0.057 ± 0.02 ^ab^	0.030 ± 0.01 ^a^
6	15	65	30	148.07± 11.10 ^b^	6.56 ± 0.23 ^b^	9.51 ± 0.99 ^a^	1.77 ± 0.38 ^abc^	2.49 ± 0.14 ^e^	23.55 ± 4.27 ^bc^	0.063 ± 0.02 ^b^	0.083± 0.01 ^b^	0.050 ± 0.00 ^bc^
7	7.5	65	15	134.48 ± 12.04 ^b^	5.51 ± 0.31 ^b^	10.60 ± 1.06 ^a^	1.37 ± 0.76 ^a^	2.30 ± 0.14 ^de^	21.26 ± 1.02 ^a^	0.040 ± 0.00 ^a^	0.063 ± 0.00 ^ab^	0.043 ± 0.00 ^ab^
8	7.5	40	30	143.23 ± 6.96 ^b^	4.05 ± 0.16 ^a^	13.46 ± 1.34 ^b^	2.24 ± 0.14 ^bc^	2.34 ± 0.17 ^de^	27.53 ± 1.15 ^bc^	0.107 ± 0.02 ^c^	0.083 ± 0.00 ^b^	0.050 ± 0.01 ^bc^
9	0	40	15	93.52 ± 3.58 ^a^	5.98 ± 0.18 ^b^	10.12 ± 0.36 ^a^	1.42 ± 0.04 ^a^	2.16 ± 0.08 ^cd^	18.75 ± 0.20 ^a^	0.083 ± 0.01 ^bc^	0.053 ± 0.01 ^bc^	0.043 ± 0.02 ^ab^

Data represented as polyphenols of green rooibos crude plant extracts expressed as mean ± standard deviation (*n* = 3). ANOVA and Duncan’s multiple range tests were performed. ^a–e^ Means with different letter superscripts within and between columns denotes significant differences (*p* < 0.05).

## Data Availability

Data from this project/study is stored in the Cape Peninsula University of Technology institutional repository and will be available on request.

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
