# Peer review of "Optimising the Polyphenolic Content and Antioxidant Activity of Green Rooibos (Aspalathus linearis) Using Beta-Cyclodextrin Assisted Extraction"

_molecules, 2022, doi:10.3390/molecules27113556_

Round 1

Reviewer 1 Report

09 May -2022

Journal: Molecules

Title: Optimising the polyphenolic content and antioxidant activity of green rooibos (Aspalathus linearis) using beta-cyclodextrin assisted extraction

Dear Editor:

The authors have evaluated the antioxidant activity and optimization of β-cyclodextrin (β-CD)-assisted extraction of crude green rooibos (Aspalathus linearis). The manuscript carries the scientific merit and would be suitable for publication after major revision as suggested below.

Nermeen Yosri, PhD

Comments to authors:

  • The graphical abstract is highly recommended.
  • " polyphenol extraction from green rooibos." Should be "polyphenol content of green rooibos."
  • Please add a dose of β-cyclodextrin (β-CD) which was used for extraction
  • " better known"; could you replace better by traditionally
  • "grows naturally only in the Fynbos biome"; please double-check and omit only
  • "an invaluable traditional medicine"; could you add examples
  • "[2]–[5]; please double check
  • "with bio-functionality are phenolic compounds" please mention examples
  • "In addition, β-CD-assisted extraction also resulted in higher content of polyphenols than hydro ethanol"; please add a reference
  • " heating the mixture at 40 – 90°C"; this is very a high temperature and may be responsible for the degradation of some compounds
  • "0.5 mg.mL-1" should be "5 mg.mL-1"; please apply for the all
  • Please add some photos of experiments to the material and methods section?
  • Could you explain table 1 in more details?
  • The authors would add the originality of the work.
  • Keywords should be more specified.
  • The discussion part could be more focused.
  • The conclusion section should be reduced with focusing on the main outcomes of the study.
  • Could you update the list of references; some of the references are more than 10 years?
  • The authors could benefit from the following reference:

El-Seedi, H.R., Yosri, N., Khalifa, S.A., Guo, Z., Musharraf, S.G., Xiao, J., Saeed, A., Du, M., Khatib, A., Abdel-Daim, M.M. and Efferth, T., 2021. Exploring natural products-based cancer therapeutics derived from Egyptian flora. Journal of Ethnopharmacology, 269, p.113626.

Taken together:

  • The authors would unify the style of the references based on the journal instructions.
  • English editing is highly required.
  • Please, re-check the punctuations, syntax, and English grammar throughout the manuscript.

Author Response

Dear reviewer #1

Thank you for carefully reading our manuscript. All the comments and suggestions received in this study have been taken into account when improving the quality of the article, and we present our reply to each of them.

Reviewer #1

  • The graphical abstract is highly recommended.

The graphical abstract was produced, and a technical error might have occurred that it was omitted while uploading the required documents.  To prove this, please see the figure below (attached document) of what was produced by the authors.

  • " Polyphenol extraction from green rooibos." Should be "polyphenol content of green rooibos."

Corrected see line 38

  • Please add a dose of β-cyclodextrin (β-CD) which was used for extraction

The two doses were 7.5 and 15 Mm (β-CD, translated to 0.85 and 1.7%. This was then stated in the methodology section, lines 138 - 140. The mM was selected since the aqueous solution was used.

  • " Better known"; could you replace better by traditionally?

Corrected see line 43

  • "Grows naturally only in the Fynbos biome"; please double-check and omit only

Corrected see line 44

  • "An invaluable traditional medicine"; could you add examples

Corrected. We added examples of the traditional use of rooibos reported by indigenous people of the Cederberg area. See lines 46 -49.

  • "[2]–[5]; please double check

Corrected, all the citations were corrected to either [2,5] or [2-5] where necessary, see lines 51, 59 etc

  • "With bio-functionality are phenolic compounds" please mention examples

Corrected see lines 56-58

  • "In addition, β-CD-assisted extraction also resulted in higher content of polyphenols than hydro ethanol"; please add a reference

Corrected, see lines 96 and 100 authors [20] Rajha et al., 2015

  • " Heating the mixture at 40 – 90°C"; this is very a high temperature and may be responsible for the degradation of some compounds

We agree with the comment made regarding the severity of these parameters. The temperatures of 40, 60 and 90°C were selected based on those applied by various studies on rooibos. The most common combination is 90-100 °C for 30 min. Prolonged heating times were selected to prove the efficiency and stability of β-CD.

  • "0.5 mg.mL-1" should be "5 mg.mL-1"; please apply for the all

Corrected, this can be seen throughout the manuscript, we selected the 0.5 mg.mL-1 form, for example see lines 30, 31, 34 etc.

  • Please add some photos of experiments to the material and methods section?

Unfortunately, we do not have any photos of the experiments besides those used in the graphic abstract

  • Could you explain table 1 in more details?

Table 1 depicts the experimental design and responses obtained for each of the nine runs. We introduced and explained what Table 1 entails from lines 234 to 243, and since the antioxidant results are depicted there, Table 1 is referred to throughout the results section. We feel that Table 1 has been explained adequately and hope the reviewer can look at it from this point.

  • The authors would add the originality of the work.

We explained the originality and reasons for undertaking the work in the cover letter.

  • Keywords should be more specified.

We battled to understand the requirement for this correction, however, we made some modification. Hoping this will be sufficient, see line 40-41

  • The discussion part could be more focused.

This paper is a section (part) of a bigger project aimed at “Evaluating the anti-glycation effect of β-CD extracted green rooibos” The rooibos plant is well researched in South Africa; however, extraction using β-CD has never been explored with rooibos. Therefore, emphasis was placed on comparing our results and that obtained by other authors in this niche. The rooibos niche is highly contested, and one must always prove that their current study is novel and improves what has been done. However, we also feel that we have explained enough the current study results.

  • The conclusion section should be reduced with focusing on the main outcomes of the study.

We acknowledge the shortcoming in this section. Instead of focusing merely on the obtained results, we compared the current study’s results with our previous work. Please see the modified section from line 630.

  • Could you update the list of references; some of the references are more than 10 years?

Since the study is based on rooibos, most of the antioxidant properties were established or most researched. Recent publications on this niche focus on quality control, microbiology, and value addition in food and cosmetics. However, we managed to change a few. Only four (4) of the 38 references are more than ten years old. 

  • The authors could benefit from the following reference:

El-Seedi, H.R., Yosri, N., Khalifa, S.A., Guo, Z., Musharraf, S.G., Xiao, J., Saeed, A., Du, M., Khatib, A., Abdel-Daim, M.M. and Efferth, T., 2021. Exploring natural products-based cancer therapeutics derived from Egyptian flora. Journal of Ethnopharmacology, 269, p.113626.

Thank you for the suggestion, although our paper was focusing on rooibos. The concept of bio-functionality of plant extracts and the effect of the solvent of extraction applied to our study, and we, therefore, included the suggested paper in our citations and references.

Taken together:

  • The authors would unify the style of the references based on the journal instructions.

We modified the manuscript and ensured that all citations appeared on the reference list, and that the style fits what is required by the Journal. Please see manuscript

  • English editing is highly required.

The paper was edited using Grammarly; please see the changed manuscript and attached report.

  • Please, re-check the punctuations, syntax, and English grammar throughout the manuscript.

Done as part of the English language editing

Reviewer 2 Report

The publication is well written, I think it is interesting and will be important for those extracting flavonoids from plants. The research was done properly, the conclusions are supported by the results. I have two minor comments:

  1. as the work concern extraction, please show an example chromatogram for a mixture of standards and extract (for the best extraction conditions)
  2. do the authors think that their work is universal? Is the method and its conditions apply only to one kind of sample? I ask that the authors try to present results for at least one other plant.

Author Response

Dear Reviewer #2

We thank you for your efforts and positive comments towards improving our manuscript. We addressed each concern to the best of our ability and hope you will be satisfied with our corrections.

Review 2#

The publication is well written, I think it is interesting and will be important for those extracting flavonoids from plants. The research was done properly, the conclusions are supported by the results. I have two minor comments:

1. As the work concern extraction, please show an example chromatogram for a mixture of standards and extract (for the best extraction conditions)

Please see the new Figure 3A & B depicting chromatograms of selected flavonoids.

2. Do the authors think that their work is universal? Is the method and its conditions apply only to one kind of sample? I ask that the authors try to present results for at least one other plant.

Yes, we do; part of this work was done in collaboration with colleagues from Argentina. They brought much knowledge and experience regarding extraction using cyclodextrins. Moreover, some of the parameters we selected matched parameters used in the rooibos research and those applied in cyclodextrin related work. For instance, method of extraction, β-CD concentration, and antioxidant activity assays are generally used across the Food Science field. One must apply what fits the sample type. For instance, we opted for the ABTS radical scavenging assay instead of DPPH, since the former was more suitable for water-soluble analytes.

Regarding presenting results for another plant, this paper is a section (part) of a bigger project aimed at “evaluating the anti-glycation effect of β-CD extracted green rooibos” thus, the whole project focused on one plant, being the green rooibos. However, we have done work in 2020 regarding green pepper. Moreover, in our discussion section, we referred to other authors who studied β-CD extraction of thyme, hemp oil, pomegranate and vine shoot to illustrate the effect of β-CD extraction.

Round 2

Reviewer 1 Report

Dear Editor

Yes, it has been modified according to our suggestions.

I recommend the paper for publication.

Kindest regards, Nermeen

Author Response

We will abide to the comment and suggestion made.